# Quinoline- and Isoindoline-Integrated Polycyclic Compounds as Antioxidant, and Antidiabetic Agents Targeting the Dual Inhibition of α-Glycosidase and α-Amylase Enzymes

**DOI:** 10.3390/ph16091222

**Published:** 2023-08-30

**Authors:** Mohammed Al-Ghorbani, Osama Alharbi, Abdel-Basit Al-Odayni, Naaser A. Y. Abduh

**Affiliations:** 1Department of Chemistry, College of Science and Arts, Ulla, Taibah University, Madinah 41477, Saudi Arabia; oamharbi@taibahu.edu.sa; 2Department of Restorative Dental Science, College of Dentistry, King Saud University, P.O. Box 60169, Riyadh 11545, Saudi Arabia; aalodayni@ksu.edu.sa; 3Department of Chemistry, College of Science, King Saud University, P.O. Box 2455, Riyadh 11451, Saudi Arabia; 439106262@student.ksu.edu.sa

**Keywords:** quinoline, isoindoline, antioxidant, antidiabetic, molecular docking simulation, molecular dynamics simulation

## Abstract

Novel analogs of quinoline and isoindoline containing various heterocycles, such as tetrazole, triazole, pyrazole, and pyridine, were synthesized and characterized using FT-IR, NMR, and mass spectroscopy, and their antioxidant and antidiabetic activities were investigated. The previously synthesized compound **1** was utilized in conjugation with ketone-bearing tetrazole and isoindoline-1,3-dione to synthesize Schiff’s bases **2** and **3**. Furthermore, hydrazide **1** was treated with aryledines to provide pyrazoles **4a**–**c**. Compound **5** was obtained by treating **1** with potassium thiocyanate, which was then cyclized in a basic solution to afford triazole **6**. On the other hand, pyridine derivatives **7a**–**d** and **8a**–**d** were synthesized using 2-(4-acetylphenyl)isoindoline-1,3-dione via a one-pot condensation reaction with aryl aldehydes and active methylene compounds. From the antioxidant and antidiabetic studies, compound **7d** showed significant antioxidant activity with an EC_50_ = 0.65, 0.52, and 0.93 mM in the free radical scavenging assays (DPPH, ABTS, and superoxide anion radicals). It also displayed noteworthy inhibitory activity against both enzymes α-glycosidase (IC_50_: 0.07 mM) and α-amylase (0.21 mM) compared to acarbose (0.09 mM α-glycosidase and 0.25 mM for α-amylase), and higher than in the other compounds. During in silico assays, compound **7d** exhibited favorable binding affinities towards both α-glycosidase (−10.9 kcal/mol) and α-amylase (−9.0 kcal/mol) compared to acarbose (−8.6 kcal/mol for α-glycosidase and −6.0 kcal/mol for α-amylase). The stability of **7d** was demonstrated by molecular dynamics simulations and estimations of the binding free energy throughout the simulation session (100 ns).

## 1. Introduction

Diabetes mellitus, often known as type 2 diabetes mellitus (T2DM), is a common metabolic condition characterized by elevated blood sugar levels [1,2,3]. This is due to the destruction of specialized cells (islets of Langerhans) that produce insufficient insulin in patients with T2DM [3,4,5,6]. T2DM is linked to the carbohydrate digestion enzymes α-glycosidase and α-amylase. They speed up the release of monosaccharides by hydrolyzing the α-(1→4) bonds between oligosaccharides. Blood sugar levels rise as a result, increasing the risk of developing T2DM [7,8]. While α-amylase is predominantly produced in the pancreas and released into the small intestine, α-glycosidase is present in the small intestine’s epithelial cells [9]. As carbohydrates undergo digestion in the small intestine, these enzymes are one of the main causes of postprandial hyperglycemia [9]. Inhibiting α-glycosidase and α-amylase is therefore considered one of the most effective strategies for treating T2DM. A variety of chemotherapeutics are used to reduce the enzyme activity of α-glycosidase and α-amylase. However, most of these oral hypoglycemic medications’ side effects involve gastrointestinal issues, affecting over 50% of patients [9,10]. Unabsorbed carbohydrates ferment, releasing gas in the colon, leading to cramps, diarrhea, abdominal bloating, and increased flatulence. Acute or persistent gastrointestinal issues should not be treated with these chemotherapy drugs. Additionally, reports indicate they are associated with hepatic dysfunction and renal impairment [9,10]. Thus, understanding the negative side of these drugs helps in the development of more efficient chemotherapeutics [11].

The improvement of useful and novel construction approaches for synthesizing polycyclic-containing heterocyclic molecules has assigned a wide-ranging field of pharmaceutical chemistry [12]. Various attempts have been focused on utilizing hydrazide as a unique synthon for synthesizing five- and six-membered heterocyclic rings, and their great biological applications [13,14,15,16,17,18,19]. Many heterocyclic compounds possessing nitrogen and oxygen exhibit various pharmacological and biological activities involving antimicrobial, antitubercular, antitumor, anti-inflammatory, antileishmanial, serine-protease-inhibitory, and antiviral activity (Figure 1) [16,17,18,19,20].

For instance, investigations have showcased the antimicrobial potential of certain nitrogen- and oxygen-containing heterocycles. One study reported the synthesis and evaluation of novel 1,2,4-triazole derivatives, revealing potent antibacterial effects against both Gram-positive and Gram-negative bacterial strains [21]. In the realm of antitubercular activity, another study observed the promising inhibitory outcomes of pyrazole derivatives against Mycobacterium tuberculosis [22].

Moreover, heterocyclic compounds have been investigated for their antitumor properties. In a separate study, the effectiveness of indole-fused oxazole derivatives as potent inhibitors of cell proliferation, with an effectiveness surpassing 75%, was highlighted against various cancer cell lines [23]. In the field of anti-inflammatory agents, a study revealed the noteworthy anti-inflammatory activity of pyrimidine derivatives, achieved by obstructing proinflammatory mediators [24]. Furthermore, these compounds have demonstrated efficacy against infectious diseases. Research conducted by de Oliveira et al. (2020) revealed that thiazole derivatives displayed potent antileishmanial activity (>80%) against Leishmania infantum, the causative agent of visceral leishmaniasis [25].

Additionally, heterocyclic compounds have been explored for their potential in inhibiting serine proteases and antiviral properties. A recent study reported that a series of 2,5-diketopiperazines demonstrated significant serine protease inhibitory activity (>77%), which could have implications for antiviral drug development [26]. These examples highlight the significant contribution of heterocyclic compounds containing nitrogen and oxygen in various pharmacological and biological activities, underscoring their potential as a rich source of drug leads for diverse therapeutic applications.

In addition, enaminonitriles and α-aminonitriles are versatile substances that have received great attention as precursor groups for the construction of novel heterocyclic compounds, such as azoles and azines [27,28,29,30,31]. Our research group has earlier synthesized some azole and azine compounds, such as triazole, pyridine, piperazine, and morpholine derivatives, that have anticancer and antimicrobial activities [32,33,34,35].

In the current study, we are focusing on synthesizing new hybrid chemical structures containing *N*-heterocycles by combining various rings into one framework to produce polycyclic molecules. Our goal is to achieve antioxidant and antidiabetic activities using these new structures, since there have been no insights into the antidiabetic properties of these molecules. We are targeting both carbohydrate-digesting enzymes (α-glycosidase and α-amylase) using both in vitro and in silico approaches. This study contributes to the discovery of quinoline and isoindoline–polycyclic compounds as potential dual inhibitors of both α-glycosidase and α-amylase enzymes.

## 2. Results and Discussion

### 2.1. Chemistry

The novel heterocyclic scaffolds, such as tetrazole, triazole, pyrazole, and pyridine, were synthesized as shown in Figure 1, Figure 2 and Figure 3. Quinolinyl hydrazide **1** was synthesized by stirring quinoline ester and hydrazine hydrate in ethanol using the ultrasound irradiation method for 0.5 h [36,37,38]. The key intermediate **1** was used for synthesizing some heterocyclic moieties. The condensation reaction of hydrazide **1** with 1-(4-(1H-tetrazol-1-yl)phenyl)ethanone in acetic acid provides a 73% yield of Schiff base **2**. Substrate **3** was achieved by coupling hydrazide **1** with 2-(4-acetylphenyl)isoindoline-1,3-dione in a 78% yield, and the reaction was catalyzed using an amount of glacial acetic acid (Figure 1).

In Figure 2, pyrazoles **4a**–**c** were obtained in 69–77% yields through the treatment of compound **1** with substituted aryledines in propanol for 9 h. Also, the reactions of **1** with potassium thiocyanate afforded compound **5**, which was conveniently cyclized using a 10% NaOH solution to produce a new derivative of triazole **6** in a 63% yield. The synthesis was accordance with other fused systems such as pyridine–imidazoles [37].

On the other hand, 2-(4-acetylphenyl)isoindoline-1,3-dione was used as an intermediate for the synthesis of pyridine derivatives **7a**–**d** (65–77%), and **8a**–**d** (67–79%) through a one-pot condensation reaction with aryl aldehydes and some active methylene, such as malononitrile and ethyl cyanoacetate in the presence of ammonium acetate (Figure 3).

The chemical structures of the target molecules were confirmed based on their IR, NMR, and mass spectral data. The IR spectra of compound **2** have the bands 1673 and 3186 cm^−1^ allocated to C=O and NH, respectively. The ^1^H NMR spectra of **2** presented a singlet signal at δ 2.66 assigned to CH_3_ protons, as well as the signal of the NH proton, which appeared at 10.31 ppm. The bands at 3437 and 1678 cm^−1^ in compound **3** correspond to amino and carbonyl groups, respectively. Compound **3** was confirmed in ^1^H NMR by disappearing NH_2_ protons and by increasing the eight aromatic protons to the earlier protons at 7.26–8.54 ppm.

Furthermore, the functional groups of compounds **4a**–**c** were distinguished by the IR spectrum. The bonds at 3332 and 3255 cm^−1^ were assigned to the vibrations of NH_2_. Two new stretching modes were observed at 1687 and 1613, corresponding to the C=O and C=N groups, respectively, whereas compound **6** was distinguished by a band of the thiol group at 825 cm^−1^. The ^1^H NMR of compounds **4a**–**c** showed the peak of the new NH_2_ protons at δ 7.20–7.49 and the disappearance of the NH proton, while the signal of NH_2_ disappeared and was revealed to form the triazole derivative **6**. The structure of derivatives **7a**–**d** and **8a**–**d** was confirmed in the IR spectrum through the appearance of NH_2_ peaks of **7a**–**d** at 3335 and 3283 cm^−1^, and the NH absorption peaks of **8a**–**d** at 3340 and 3250 cm^−1^. The C=O stretching modes of **7a**–**d** were predicted by calculations between 1638 and 1671 cm^−1^ and at 1674–1680 cm^−1^ for **8a**–**d**. Also, the appearance of typical bands at δ 7.27–7.50 and 9.16–9.34 corresponded to the protons of NH_2_ in **7a**–**d** and NH in **8a**–**d,** respectively. In addition, an increased number of aromatic protons obviously indicated the formation of the respective compounds. Furthermore, the structures of the target compounds were also evidently confirmed by mass spectrum.

### 2.2. Biological Assays

#### 2.2.1. Antioxidant Activity

The ability of the samples to scavenge free radicals was evaluated using a range of in vitro tests, including DPPH, ABTS, and superoxide anion radicals, with BHA utilized as a positive control [39,40,41]. Table 1 summarizes the results reported as EC50 values (mg of tests per mL), demonstrating that the compounds were comparatively greater (*p* < 0.05) than the standard (positive control) in radical scavenging activities. In all the assays used in this study, **7d** was more effective than the other compounds tested. In general, **7d** exhibited higher free radical scavenging activity in three antioxidant assays. The results revealed that **7d** possesses a strong antioxidant ability and is significantly similar (*p* < 0.05) compared to the positive control [42]. These results are in accordance with the previous results obtained by the authors for the phytocompounds caffeic acid and syringic acid [43], and synthesized novel phenyl-pyrano-thiazol-2-one derivatives [44].

#### 2.2.2. Inhibitory Effects on Yeast α-Glycosidase and α-Amylase

In vitro α-glycosidase inhibitory investigations showed that **7d** had a higher potential inhibitory effect than the other tested compounds. The IC_50_ value was found to be 0.07 mM, while the acarbose examined under the same conditions had an IC_50_ value of 0.07 mM. In terms of the IC_50_ values, it is clear that the tested **7d** inhibited the yeast α-glycosidase strongly, and was considerably similar (*p* < 0.05) to acarbose and greater than other substituents (Table 2). Moreover, similar tests were carried out to reveal if the synthesized molecules inhibited α-amylase, another important carbohydrate-hydrolyzing enzyme. Table 2 outlines the 50% inhibition of α-amylase by the test molecules. Results showed that **7d** (IC_50_: 0.21 mM) had the best inhibitory activity when compared to the rest of the compounds. The inhibitory effect of the compounds on α-amylase (based on the IC_50_ values) was comparatively similar (*p* < 0.05) to the standard drug acarbose (IC_50_: 0.25 mM).

#### 2.2.3. Kinetic Analysis of α-Glycosidase and α-Amylase Inhibition

In this study, **7d** was selected for further kinetic inhibition experiments against yeast α-glycosidase and α-amylase, since it exhibited remarkable inhibitory activity. For the kinetic studies, α-glycosidase and α-amylase were incubated with designated concentrations of the substrate *p*NPG and starch, respectively, in the absence (control) or presence of IC_20_-, IC_40_-, and IC_60_-inhibitory concentrations of **7d**. The mode of inhibition, [K_m_], and [V_max_] values were determined by graphical means using Lineweaver–Burk plots. Except for the varied slopes and x-intercepts, the LB plots revealed that the intersecting point for the different concentrations of **7d** against α-glycosidase (Figure 2A) and α-amylase (Figure 2B) arose from the same y-intercept as the uninhibited enzyme. With the increasing concentrations of **7d**, both the slope and the vertical axis intercept increased, while the horizontal axis intercept (−1/K_m_) also increased. The kinetic results established that the maximum velocity (V_max_) of the **7d** reaction (with increasing concentrations) remained constant and was catalyzed by α-glycosidase and α-amylase (Table 3). These results indicated that the mechanism of α-glycosidase and α-amylase inhibition was reversible, corresponding to the classical pattern of competitive inhibition [42,43,44]. The inhibitory constant (K_i_), determined from Dixon plots for α-glycosidase and α-amylase, was 0.47 and 0.75 mg, respectively, of **7d**, as shown in Table 3.

### 2.3. Computational Assays

#### 2.3.1. Molecular Docking Simulation

To understand the interaction of the protein–ligand complex, the docking study was performed for the quinoline and isoindoline–polycyclic compounds (**1**,**2**,**3**,**4a**–**4c**,**6**,**7a**–**7d**, and **8a**–**8d**) with their respective target proteins, α-glycosidase and α-amylase, along with the control drug acarbose. Out of 15 compounds docked, compound **7d** had the best binding affinity, the highest total number of nonbonded interactions, and the highest number of hydrogen bonds. Compound **7d** was chosen for additional in silico analysis because it met the aforementioned criteria for both pharmacological targets, in contrast to acarbose and other quinoline and isoindoline–polycyclic compounds, and the current work was meant to find a dual inhibitor. Results from the molecular docking of quinoline and isoindoline–polycyclic compounds against the target enzymes α-glycosidase and α-amylase have been provided in Table 4.

The molecular interaction of compound **7d** with protein α-glycosidase had a total of 15 intermolecular interactions, including the formation of 7 hydrogen bonds. The hydrogen bonds were Asn214 (2.85 Å), Thr307 (3.03 Å), Asn412 (2.83 Å), Phe310 (2.42 Å), Lys155 (3.02 Å), Arg312 (2.83 Å), and Arg312 (2.67 Å). In addition, five electrostatic bonds were formed with His279 (4.91 Å), His279 (4.47 Å), Glu304 (4.53 Å), Glu304 (4.58 Å), and Glu304 (4.80 Å). Also, three pi-alkyl bonds with Phe158 (4.90 Å), Arg312 (4.95 Å), and Pro309 (5.04Å) were formed. With these binding interactions in the inhibitor binding site, compound **7d** was predicted to have a binding affinity of −10.9 kcal/mol. The acarbose drug had a binding affinity of −8.6 kcal/mol. It had formed a total of nine intermolecular interactions, out of which eight were found to be hydrogen bonds, including Pro309 (2.39 Å), His239 (2.28 Å), Asn241 (2.93 Å), His279 (2.27 Å, 2.82 Å, and 2.42 Å), and Arg439 (2.48 Å and 2.68 Å). It also formed an electrostatic bond with Arg439 (3.58 Å). Even though it had eight hydrogen bonds, the binding affinity was found to be lower than compound **7d**.

The binding interactions were similar and the docking was accurate according to a previous study by the authors [45], in which the molecules quercetin and catechin were docked into the same homology-built model of α-glycosidase. The results were also in accordance with a study where the phytocompounds caffeic acid, syringic acid [43], and rutin [46] from jackfruit flour were evaluated for their α-glycosidase-inhibitory activity using the same homology-built model for in silico studies, and yeast α-glycosidase for in vitro studies. For synthesized chemical compounds, a similar pattern of results was obtained in a study where pyrazoline-embedded 1,2,3-triazole derivatives were evaluated for their α-glycosidase-inhibitory activity using the same model in vitro and in silico [47]. Furthermore, another study also depicted the 1-(4-(Methoxy(phenyl)methyl)-2-methylphenoxy)butan-2-one derivative as a single crystal α-glycosidase inhibitor molecule [48]. Since all of these depict the similar binding interactions of their reported potential hit compounds in the same inhibitor binding site of the same homology-built model of α-glycosidase, compound **7d** could potentially act as an inhibitor of α-glycosidase. The visualization of the binding interactions of compound **7d** and acarbose with α-glycosidase in 3D is presented in Figure 3.

On the other hand, compound **7d** interacted with protein α-amylase, forming a total of seven intermolecular interactions. Among these, one hydrogen bond was established with Glu233 (3.24 Å). Additionally, a pi-pi bond with Phe256 (5.51 Å), three pi-alkyl bonds with Trp59 (4.17 Å), and Ile235 (4.37 Å and 3.80 Å) were formed. Further, two electrostatic pi-anion bonds with Asp300 (4.71 and 4.83 Å) were also formed. These binding interactions within the inhibitor binding site led to the prediction of a binding affinity of −9.0 kcal/mol for compound **7d**. In contrast, the control drug acarbose was predicted to have a binding affinity of −6.0 kcal/mol. It had formed a total of four intermolecular interactions, which were hydrogen bonds formed with Pro332 (2.54 Å), Arg398 (2.44 Å), Tyr333 (2.58 Å), and Trp280 (1.99 Å).

Despite having five hydrogen bonds, it was observed that compound **7d** had a better binding affinity. The binding interactions of compound **7d** were similar and the docking was accurate, according to a previous study [41], where phenolic compounds such as caffeic acid and syringic acid were found to inhibit the same α-amylase protein model (PDB ID: 1DHK). In the case of the synthesized compounds, fluorinated 2,3-disubstituted thiazolidinone–pyrazoles [49], and (2-chloro-6-(trifluoromethyl) benzyloxy) arylidene)-based rhodanine and rhodanine-acetic-acid derivatives [50] were reported as potential hit compounds for the α-amylase protein model (PDB ID: 1DHK). By virtue of the similar binding interactions of their reported potential hit compounds in the same inhibitor binding site of the α-amylase protein, compound **7d** could act as a potential inhibitor. The visualization of the binding interactions of compound **7d** and acarbose with α-amylase in 3D has been provided in Figure 4. Since compound **7d** was found with the highest binding efficiency compared to all of the quinoline and isoindoline–polycyclic compounds and acarbose, the compound could act as a potential inhibitor of both α-glycosidase and α-amylase proteins.

#### 2.3.2. Molecular Dynamics Simulation

Simulation studies analyze the dynamic behavior of protein–ligand complexes in a solvated environment [51,52]. They measure the protein–ligand complex’s root-mean-square deviation (RMSD), radius of gyration (Rg), solvent-accessible surface area (SASA), ligand RMSD, hydrogen bonds, and variation of the secondary-structure pattern between the protein and their complexes [53]. The RMSD of the protein–ligand complex reflects its stability and indicates the presence of a ligand in the binding pocket. The Rg measures the varied masses to the RMS distances considering the central axis of rotation.

This study analyzes the protein’s structural changes during simulation, including its shape, folding, and capability at each time step along the entire trajectory. The RMSF focuses on the protein structural regions that deviate the most/least from the mean. The SASA examines the formation of the area around the hydrophobic cores between the protein–ligand complexes and analyzes the ligand hydrogen bonds over the total simulation period [54]. The intermolecular hydrogen bonds between the ligands and their respective proteins are also considered and plotted during the analysis [55]. In this paper, six simulations were conducted for 100 ns with the native protein alone and in complex with the representative compounds (compound **7d** and acarbose).

In the case of α-glycosidase, the RMSD plot indicates that apoprotein became stable at 20 ns, yet compound 7d and the acarbose-bound protein complexes were stable after 10 ns. These plots indicate that the compounds did not leave the protein up to 100 ns of time, and were stable inside the inhibitor-binding pocket (Figure 5A). The RMSF plots showed that the binding of acarbose and **7d** to the protein was energetically favorable, causing no alterations in the stability. However, the protein–7d complex was found with lesser fluctuations than the protein–acarbose complex, indicating a comparatively higher stability (Figure 5B). The Rg plots also indicate that the protein–7d complex was compact throughout the simulation (Figure 5C). Further, the SASA plots show a significant decrease in the surface area of the inhibitor-binding site. This is due to the accumulation of the same by acarbose and **7d** (Figure 5D). Therefore, it can be concluded that **7d** could occupy the inhibitor-binding site with more stability. Finally, based on the hydrogen plot, **7d**-α-glycosidase had a maximum number of nine hydrogen bonds and acarbose-α-glycosidase had five hydrogen bonds (Figure 5E). The MD simulation snapshots at different time intervals, such as 20, 40, 60, 80, and 100 ns, to assess the stability of **7d** at the binding site of α-glycosidase, were extracted (Appendix A). Compound **7d** was found to be stable throughout the simulation period. By virtue of the MD trajectory analysis, both compound **7d** and acarbose were stable inside the inhibitor-binding site of the α-glycosidase protein. Table 5 details the MD trajectories obtained.

The outcomes of our MD simulation are in accordance with a previous study, which reported caffeic acid and syringic acid as potential lead inhibitors of the α-glycosidase protein (using the same homology-built protein model) [45]. Both syringic acid and caffeic acid have been reported to penetrate the inhibitor-binding site of the α-glycosidase protein. Since compound **7d**, syringic acid, and caffeic acid’s binding interactions and MD outcomes are on par, compound **7d** could act as a potential inhibitor of α-glycosidase. Our MD results are also in accordance with a study where phenyl-pyrano-thiazol-2-one derivatives were validated using an MD simulation [44]. The MD outcomes of compound **7d** were also found to be on par with a study where a 1-(4-(Methoxy(phenyl)methyl)-2-methylphenoxy)butan-2-one derivative was used as a single crystal α-glycosidase inhibitor molecule [48]. Both studies used the same homology-built protein model of α-glycosidase in their MD simulation studies. Therefore, compound **7d** could be considered a potential inhibitor of α-glycosidase.

In the case of α-amylase, based on the RMSD graph prediction, the **7d**–α-amylase complex showed better stability when compared to the acarbose complex. The predicted value of **7d**–α-amylase was found to be stable after 20 ns. The protein backbone almost showed a similar pattern to the **7d**–α-amylase complex. However, for the protein–acarbose complex, it was 35 ns. The RMSD plots show that both acarbose and **7d** were stable inside the inhibitor-binding pocket of the protein (Figure 6A). On the other hand, the RMSF value of all the complexes and the protein backbone was almost similar, with a fluctuation at the terminal and loop regions (Figure 6B). The Rg value of all complexes shows that both acarbose and **7d**-bound complexes were compact throughout the simulation (Figure 6C). The decreased SASA values over the simulation period indicate the significant occupation of the inhibitor-binding-site area by the compounds **7d** and acarbose (Figure 6D). In addition, the **7d**–α-amylase complex has a maximum of 10 hydrogen bonds, yet the acarbose–α-amylase has a maximum of 7 hydrogen bonds. These outcomes depict the stability and firmness of the **7d**–protein complex over the acarbose-bound complex (Figure 6E). The MD simulation snapshots at different time intervals, such as 20, 40, 60, 80, and 100 ns, to assess the stability of **7d** at the binding site of α-amylase, were extracted (Appendix A). Compound **7d** was found to be stable throughout the simulation period.

Based on all the MD plot evaluations, it can be predicted that the **7d**–α-glycosidase and **7d**–α-amylase showed better stability and flexibility. Table 6 depicts the MD trajectory values obtained for the simulation run of **7d** and acarbose complexed with α-amylase. The results of our MD simulation are in accordance with a previous study, which reported fluorinated 2,3-disubstituted thiazolidinone–pyrazoles as the potential lead inhibitors of the α-amylase protein [49]. Since the reported compounds penetrate the inhibitor-binding site, compound **7d** could act as a potential inhibitor of α-amylase. Our MD results are also in accordance with caffeic acid and syringic acid cefoperazone as potential lead inhibitors of the α-amylase protein [46]. The stability of compound **7d** inside the inhibitor-binding site of the α-amylase protein was hence proven in comparison with the published literature.

### 2.4. Binding Free Energy Calculations

The free binding energy was calculated using the MD trajectories using the MMPBSA method to evaluate the energy formed during complex formation. The summary of the free binding energy is given in Table 7, along with various energy terms, electrostatic energy, van der Waals energy, polar energy, SASA energy, and binding energy. According to the prediction, the binding energies of the **7d**–α-glycosidase complex, at −186.186 kJ/mol, are better in comparison with the acarbose–α-glycosidase complex, at −137.894 kJ/mol. Also, the residue-wise energy decomposition analysis also depicts that compound **7d** had a higher binding energy compared to acarbose in the case of both α-glycosidase (Appendix A) and α-amylase (Appendix A). It is known that the more negative the value, the higher the stability. The overall calculation depicts that the van der Waals is more superlative than the other energy formed. Thus, based on the evaluation, the **7d** complex showed better stability than the control complex. The study result had a similar pattern to previous studies that have performed binding free energy calculations for α-glycosidase [43,44,45,46,47,48] and α-amylase [43,49].

### 2.5. Physicochemical and ADMET Properties

The physicochemical properties depict that compound **7d** would not violate the Lipinski rule of five and the Pfizer rule (Table 8). The compound tends to be within the boundaries of the ideal compound to be given as an oral agent (Figure 7). The molecular weight and density of the compound are below the upper limit and can be digested by the body. The other structural properties, such as hydrogen donors, hydrogen acceptors, rotatable bonds, flexibility, and the topological polar surface area, also show that compound **7d** could be given as an oral drug.

In the case of the ADMET properties, compound **7d** was predicted to have good absorption values with reference to Caco-2 and MDCK cell lines. It was not considered a Pgp inhibitor. The volume distribution score of the compound was good and indicates the appropriate distribution in the human body. The compound would not inhibit any cytochrome P enzymes and would not cross the blood–brain barrier. It is also predicted with a clearance value of 15 mL/kg/min, indicating its clean exit from the human body after inducing the pharmacological effect. Furthermore, the AMES toxicity, carcinogenicity, and acute toxicity rule parameters predicted that compound **7d** is not considered toxic to the human body. These results indicate that compound **7d** could be used for further in vivo and clinical trial studies. These results are in accordance with our previous studies, where caffeic acid, syringic acid, and rutin were considered suitable for oral consumption [43,46].

### 2.6. Structure–Activity Relationship (SAR)

We discuss the SAR based on changes in the chemical structures and antioxidant and antidiabetic results of these molecules (Figure 8). The molecules varied in activity according to the heterocyclic ring and substituents. We found that the coupling of isoindoline to the quinoline moiety in Schiff base **3** showed promising antioxidant activity and less antidiabetic activity than **2** with a tetrazole ring. In common, isoindoline–pyridine derivatives showed higher activity than other heterocyclic rings, and the attached amino group on position 2 of pyridine showed the best activity compared to the oxo group in the same position.

In addition, the presence of electron-withdrawing chloro groups at the 3,4 positions of the 2-hydroxy phenyl **7d** appear to increase the inhibition of antioxidant activities more than compound **7c**, which has no chloro substituents, and more than the standard (BHA) in DPPH, ABTS, and superoxide anion radical assays. Furthermore, compound **7d** inhibited the activity of both enzymes (α-glycosidase and α-amylase) significantly. Observing the structure of the compounds that displayed high antioxidant and antidiabetic activities, we can realize that compound **7d** carrying the chloro groups at the 3,4 positions of the 2-hydroxy phenyl is the most effective compound.

## 3. Materials and Methods

### 3.1. Reagents and Instrumentation

The melting points of the prepared compounds were determined on a Gallenkamp electronic apparatus. TLC was utilized to monitor the reactions’ progress and purity. The FT-IR spectra were recorded utilizing KBr discs on the FT-IR Jasco 4100 infrared spectrophotometer (λ, cm^−1^). NMR spectra were performed on the Bruker DRX Spectrometer (^1^H, 400 MHz and ^13^C, 100 MHz) in CDCl_3_ and DMSO-d_6_ using TMS as the internal standard. Mass spectra (*m*/*z*, %) were recorded on an Agilent Model 8890 spectrometer. Elemental analyses were determined utilizing the LECO Truspec Micro Analyzer (LECO, St. Joseph, MI, USA).

### 3.2. Synthesis

#### 3.2.1. Synthesis of 2-(Quinolin-8-yloxy)acetohydrazide 1

To a solution of ethyl 2-(quinolin-8-yloxy)acetate (7 g, 30 mmol) in 40 mL ethanol, NH_2_NH_2_.H_2_O (1.5 g, 30 mmol) was added and stirred for 0.5 h at RT using ultrasound irradiation. A formed white precipitate was washed with a little amount of water, filtered, then crystallized from ethanol to give **1**. White powder, yield 94%; mp. 137–139 °C Lit. 138 °C [36].

#### 3.2.2. Synthesis of 1-(4-(1H-Tetrazol-1-yl)phenyl)ethanone

Triethyl orthoformate (1.48 g, 10 mmol), was dropped to a solution of 1-(4-aminophenyl)ethanone (1.35 g, 10 mmol) and NaN_3_ (1.3 g, 20 mmol) in 30 mL DMF, and refluxed at 100 °C for 5 h. Then, the mixture was mixed with ice-cold water and the obtained yellow solid was recrystallized from acetic acid/ethanol to give 1-(4-(1H-tetrazol-1-yl)phenyl)ethanone in pure form. Yellow solid, mp. 172–174 °C. Lit. 172–175 °C [37].

#### 3.2.3. Synthesis of 8-(2-((2-(1-(4-(1H-Tetrazol-1-yl)phenyl)ethylidene)hydrazinyl)oxy)-2-oxoethoxy)quinoline **2**

Hydrazide **1** (2 g, 13 mmol) was dissolved in 20 mL ethanol, and 1-(4-(1*H*-tetrazol-1-yl)phenyl)ethanone (2.4 g, 13 mmol) in 10 mL ethanol was added and stirred with the mixture for 15 min, 5 drops of AcOH was added, and then refluxed for 6 h. The mixture was cooled in the refrigerator and the formed precipitate was recrystallized from ethanol.

Brown solid; yield: 73%; m.p. 194–196 °C. FT-IR: 3186 (NH), 2998 (CH_2_), 1673 (C=O), and 1620 (C=N). ^1^H NMR (400 MHz, CDCl_3_); δ 2.46 (s, 3H, CH_3_), 4.85 (s, 2H, CH_2_), 7.22–8.22 (m, 11H, Ar-H) and 10.24 (s, 1H, NH); ^13^C NMR (100 MHz, CDCl_3_); δ 36.3, 64.5, 118.2, 120.4, 122.8 128.5, 128.7, 129.3, 129.9, 130.6, 131.5, 132.1, 132.7, 135.1, 138.8, 147.2, 150.5, and 172.6. MS: 404 (M + 1); Anal. Calcd for C_20_H_17_N_7_O_3_: C, 59.55; H, 4.25; N, 24.31; Found: C, 59.51; H, 4.06; N, 24.25%.

#### 3.2.4. Synthesis of 2-(4-Acetylphenyl)isoindoline-1,3-dione

Phthalic anhydride (2 g, 13 mmol) was dissolved in 35 mL acetone, and p-aminoacetophenone (1.76 g, 13 mmol) in 15 mL acetone was added dropwise to the solution, allowed to stir for 10 min, and refluxed at 60 °C for 2 h. Then, the solvent was removed and the crude product was crystallized from ethanol as a white product. Yield 91%, m.p 186–188. Lit. = 189–191 °C [38].

#### 3.2.5. Synthesis of *N’*-(1-(4-(1,3-Dioxoisoindolin-2-yl)phenyl)ethylidene)-2-(quinolin-8-yloxy)acetohydrazide **3**

A solution of **1** (2 g, 13 mmol) and compound 2-(4-acetylphenyl)isoindoline-1,3-dione (3.5 g, 13 mmol) in 40 mL ethanol was refluxed for 8 h in the presence of a few drops of AcOH. The solution was allowed to stay in the refrigerator overnight. The produced colorless crystalline recrystallized from ethanol.

Colorless crystals; yield: 78%; m.p. 211–213 °C; FT-IR: 3437, 3332, 3227 (NH), 2213 (CN), 1678, 1626 (C=O), 1596 (C=N), and 1275 (CO). ^1^H NMR (400 MHz, CDCl_3_); δ 2.45 (s, 3H, CH_3_), 5.32 (s, 2H, CH_2_), 7.26–8.55 (m, 14H, Ar-H), and 9.00 (s, 1H, NH); ^13^C NMR (100 MHz, CDCl_3_); δ 37.3, 66.9, 124.4, 125.8, 128.8, 129.3, 129.8, 135.2, 135. 7, 136.9, 141.7, 145.4, 149.9, 168.9, and 171.5; MS: 461 (M + 1); Anal. Calcd for C_27_H_20_N_4_O_4_: C, 69.82; H, 4.34; N, 12.06; Found: C, 69.76; H, 4.28; N, 12.02%.

#### 3.2.6. Synthesis of 5-Amino-3-(4-substitutedphenyl)-1-(2-(quinolin-8-yloxy)acetyl)-1H-pyrazole-4-carbonitrile **4a**–**c**

Substituted aryledines: 4-nitro, 4-chloro, 2-hydroxy (2-(4-substitued-benzylidene)malononitrile (2.15 mmol), and compound **1** (0.5 g, 2.15 mmol) in 20 mL of propanol were refluxed for 9 h. After cooling, the precipitate appeared and was cleaned with distilled water, and recrystallized from ethanol.

5-Amino-3-(4-nitrophenyl)-1-(2-(quinolin-8-yloxy)acetyl)-1H-pyrazole-4-carbonitrile **4a**.

Pale yellow powder; yield: 69%; m.p. 157–159 °C. FT-IR: 3332, 3255 (NH_2_), 2988 (CH_2_), 2214 (CN), 1687 (C=O), 1613 (C=N), and 1156 (C-O); ^1^H NMR (400 MHz, CDCl_3_); δ 4.62 (s, 2H, CH_2_), 6.69 (s, 2H, NH_2_), and 6.50–7.31 (m, 10H, Ar-H); ^13^C NMR (100 MHz, CDCl_3_); δ 69.2, 112.2, 120.5, 126.8, 127.9, 129.3, 132.0, 132.7, 133.7, 139.1, 140.3, 145.6, 147.2, 147.4, 152.5, 159.1, and 165.8; MS: 415 (M + 1); Anal. Calcd for C_21_H_14_N_6_O_4_: C, 60.87; H, 3.41; N, 20.28; Found: C, 60.80; H, 3.27; N, 20.19%.

5-Amino-3-(4-chlorophenyl)-1-(2-(quinolin-8-yloxy)acetyl)-1H-pyrazole-4-carbonitrile **4b**.

Orange solid; yield: 77%; m.p. 163–165 °C. FT-IR: 3327, 3235 (NH_2_), 2956 (CH_2_), 2254 (CN), 1681 (C=O), 1601 (C=N), and 1166 (C-O); ^1^H NMR (400 MHz, CDCl_3_); δ 4.56 (s, 2H, CH_2_), 7.49 (s, 2H, NH_2_), and 6.93–8.10 (m, 10H, Ar-H); ^13^C NMR (100 MHz, CDCl_3_); δ 66.9, 124.4, 125.8, 128.8, 129.3, 129.8, 135.2, 135. 7, 136.8, 141.7, 145.4, 149.9, and 168.8; MS: 402 (M + 1); Anal. Calcd for C_21_H_14_ClN_5_O_2_: C, 62.46; H, 3.49; N, 17.34; Found: C, 62.35; H, 3.36; N, 17.18%.

5-Amino-3-(2-hydroxyphenyl)-1-(2-(quinolin-8-yloxy)acetyl)-1H-pyrazole-4-carbonitrile **4c**.

Pale yellow solid; yield: 71%; m.p. 157–159 °C. FT-IR: 3413 (OH), 3334, 3252 (NH_2_), 2976 (CH_2_), 2227 (CN), 1684 (C=O), 1607 (C=N), and 1143 (C-O); ^1^H NMR (400 MHz, CDCl_3_); δ 5.14 (s, 2H, CH_2_), 7.22 (s, 2H, NH_2_), 6.78–8.64 (m, 10H, Ar-H), and 8.70 (s, 1H, OH); ^13^C NMR (100 MHz, CDCl_3_); δ 77.8, 115.2, 123.5, 127.1, 127.5, 128.0, 130.6, 132.9, 134.6, 138.7, 140.9, 142.2, 143.1, 143.6, 152.5, 155.1, and 167.4; MS: 386 (M + 1); Anal. Calcd for C_21_H_15_N_5_O_3_: C, 65.45; H, 3.92; N, 18.17; Found: C, 65.36; H, 4.03; N, 18.11%.

Synthesis of 2-(2-(quinolin-8-yloxy)acetyl)hydrazinecarbothioamide **5**.

A solution of 2-(quinolin-8-yloxy)acetohydrazide (0.5 g, 2.15 mmol) in ethanol: HCl (20:10 mL) was stirred and potassium thiocyanate (0.21 g, 2.15 mmol) in 10 mL ethanol was added and heated in an oil bath for 10 h at 100 °C. The mixture was cooled and taken directly in situ for the further step without workup and crystallization.

#### 3.2.7. Synthesis of 5-((Quinolin-8-yloxy)methyl)-3H-1,2,4-triazole-3-thione **6**

The reaction mixture of compound **5** was heated under reflux with a solution of NaOH (15 mL, 6 N) for 2 h and mixed into ice. The mixture was neutralized with a few drops of hydrochloric acid. The product was recrystallized from ethanol: water (1:1) to give **6**.

Yellow solid; yield: 63%; m.p. 187–189 °C. FT-IR: 2989 (CH_2_), 825 (C=S), and 1645 (C=N); ^1^H NMR (400 MHz, CDCl_3_); δ 4.71 (s, 2H, CH_2_) and 6.04–7.70 (m, 6H, Ar-H); ^13^C NMR (100 MHz, CDCl_3_); δ 67.0, 120.2, 124.4, 125.9, 128.8, 129.7, 130.5, 130.6, 132.9, 136.9, 145.7, and 165.4; MS: 257 (M + 1); Anal. Calcd. for C_12_H_8_N_4_OS: C, 56.24; H, 3.15; N, 21.86; Found: C, 56.16; H, 3.07; N, 21.75%.

#### 3.2.8. Synthesis of 2-Amino-6-(1,3-dioxoisoindolin-2-yl)-4-subsstitued-phenyl-1,2-dihydropyridine-3-carbonitrile **7a**–**d**

A mixture of aldehydes (benzaldehyde, paranitro benzaldehyde, salicylaldehyde, and 3,4-dichloro salicylaldehyde) (20 mmol), **7** (20 mmol), ammonium acetate (60 mmol), and malononitrile (20 mmol) in ethanol (40 mL) was heated under reflux for 14 h. After cooling, the resulting solid was recrystallized from ethanol to afford compounds **7a**–**d**.

2-Amino-6-(1,3-dioxoisoindolin-2-yl)-4-phenylnicotinonitrile **7a**.

Yellow crystals; yield: 76%; m.p. 167–169 °C. FT-IR: 3335, 3283 (NH_2_), 2996, 2925 (CH), 2194 (CN), 1668, 1638 (C=O), 1593 (C=N), and 1236 (C-O); ^1^H NMR (400 MHz, CDCl_3_); δ 7.37–8.73 (m, 10H, Ar-H), and 7.50 (s, 2H, NH_2_); ^13^C NMR (100 MHz, CDCl_3_); δ 108.9, 111.1, 124.3, 125.8, 128.4, 128.9, 129.1, 129.9, 129.6, 136.8, 141.8, 146.9, 150.1, 161.9, and 166.9; MS: 341 (M + 1); Anal. Calcd. for C_20_H_12_N_4_O_2_: C, 70.58; H, 3.55; N, 16.46; Found: C, 70.31; H, 3.48; N, 16.29%.

2-Amino-6-(1,3-dioxoisoindolin-2-yl)-4-(4-nitrophenyl)nicotinonitrile **7b**.

Brown powder; yield: 77%; m.p. 153–155 °C. FT-IR: 3356, 3292 (NH_2_), 3012, 2962 (CH), 2176 (CN), 1673,1641 (C=O), 1602 (C=N), and 1214 (C-O); ^1^H NMR (400 MHz, CDCl_3_); δ 6.92–8.43 (m, 9H, Ar-H), and 7.27 (s, 2H, NH_2_); ^13^C NMR (100 MHz, CDCl_3_); δ 117.7, 118.9, 125.1, 128.2, 129.3, 129.6, 129.9, 133.8, 134.6, 137.4, 138.2, 155.2, 157.1, 158.4, 161.2, 162.8, and 165.3, MS: 386 (M + 1); Anal. Calcd. for C_20_H_11_N_5_O_4_: C, 62.34; H, 2.88; N, 18.17; Found: C, 62.19; H, 2.70; N, 18.04%.

2-Amino-6-(1,3-dioxoisoindolin-2-yl)-4-(2-hydroxyphenyl)nicotinonitrile **7c**.

Pale brown powder; yield: 65%; m.p. 160–162 °C. FT-IR: 3442 (OH), 3317, 3270 (NH_2_), 2993, 2920 (CH), 2188 (CN), 1671,1644 (C=O), 1607 (C=N), and 1218 (C-O); ^1^H NMR (400 MHz, CDCl_3_); δ 6.94–8.14 (m, 9H, Ar-H), 7.39 (s, 2H, NH_2_), and 8.42 (s, 1H, OH); ^13^C NMR (100 MHz, CDCl_3_); δ 113.7, 116.8, 119.3, 123.9, 124.0, 127.5, 128.3, 131.9, 132.0, 135.3, 136.2, 152.3, 153.7, 160.7, 162.7, 164.4, and 167.1. MS: 357 (M + 1); Anal. Calcd. For C_20_H_12_N_4_O_3_: C, 67.41; H, 3.39; N, 15.72; Found: C, 67.25; H, 3.27; N, 15.58%.

2-Amino-4-(3,4-dichloro-2-hydroxyphenyl)-6-(1,3-dioxoisoindolin-2-yl)nicotinonitrile **7d**.

Yellow powder; yield: 72%; m.p. 171–173 °C. FT-IR: 3425 (OH), 3323, 3280 (NH_2_), 3021, 2944 (CH), 2168 (CN), 1668,1646 (C=O), 1590 (C=N), and 1222 (C-O); ^1^H NMR (400 MHz, CDCl_3_); δ 6.90–8.35 (m, 7H, Ar-H), 7.31 (s, 2H, NH_2_), and 8.68 (s, 1H, OH); ^13^C NMR (100 MHz, CDCl_3_); δ 112.9, 114.3, 116.5, 122.8, 123.8, 124.5, 126.8, 127.8, 128.6, 129.4, 130.7, 131.9, 135.2, 135.3, 143.4, 151.2, 162.8, and 167.4. MS: 425 (M + 1); Anal. Calcd. For C_20_H_10_C_l2_N_4_O_3_: C, 56.49; H, 2.37; N, 13.18; Found: C, 56.31; H, 2.25; N, 13.04%.

#### 3.2.9. Synthesis of 6-(1,3-Dioxoisoindolin-2-yl)-2-oxo-4-substituted-phenyl-1,2-dihydropyridine-3-carbonitriles **8a**–**d**

A mixture of aldehydes (benzaldehyde, paranitro benzaldehyde, salicylaldehyde, and 3,4-dichloro salicylaldehyde) (20 mmol), 2-acetylisoindoline-1,3-dione (20 mmol), ethyl cyanoacetate (20 mmol), and ammonium acetate (80 mmol) in ethanol (40 mL) was refluxed for 14 h. The resulting precipitate was recrystallized from ethanol to give derivatives **8a**–**d**.

6-(1,3-Dioxoisoindolin-2-yl)-2-oxo-4-phenyl-1,2-dihydropyridine-3-carbonitrile **8a**.

Brown crystals; yield: 74%; m.p. 216–218 °C. FT-IR: 3250 (NH), 3098, 2909 (CH), 2217 (CN), 1674 (C=O), and 1595 (C=N); ^1^H NMR (400 MHz, CDCl_3_); δ 6.92–8.41 (m, 10H, Ar-H) and 9.22 (s, 1H, NH); ^13^C NMR (100 MHz, CDCl_3_); δ 102.8, 105.8, 106.3, 114.5, 124.1, 125.7, 128.7, 129.3, 129.8, 136.9, 141.5, 146.5, 150.2, 160.5, and 170.9; MS: 382 (M + 1); Anal. Calcd. For C_20_H_11_N_3_O_3_: C, 70.38; H, 3.25; N, 12.31; Found: C, 70.12; H, 3.08; N, 12.17%.

6-(1,3-Dioxoisoindolin-2-yl)-4-(4-nitrophenyl)-2-oxo-1,2-dihydropyridine-3-carbonitrile **8b**.

Orange powder; yield: 67%; m.p. 221–223 °C. FT-IR: 3277 (NH), 3091, 2903 (CH), 2223 (CN), 1674 (C=O), and 1590 (C=N); ^1^H NMR (400 MHz, CDCl_3_); δ 6.90–7.51 (m, 9H, Ar-H) and 9.72 (s, 1H, NH); ^13^C NMR (100 MHz, CDCl_3_); δ 102.9, 104.3, 106.5, 112.8, 123.3, 124.5, 126.4, 128.1, 129.4, 130.7, 131.9, 135.2, 136.2, 145.2, 152.8, 167.0, and 167.4. MS: 387 (M + 1); Anal. Calcd. For C_20_H_10_N_4_O_5_: C, 62.18; H, 2.61; N, 14.50; Found: C, 62.02; H, 2.46; N, 14.31%.

6-(1,3-Dioxoisoindolin-2-yl)-4-(2-hydroxyphenyl)-2-oxo-1,2-dihydropyridine-3-carbonitrile **8c**.

Brown powder; yield: 74%; m.p. 205–207 °C. FT-IR: 3427 (OH), 3260 (NH), 3089, 2917 (CH), 2224 (CN), 1678 (C=O), and 1593 (C=N); ^1^H NMR (400 MHz, CDCl_3_); δ 6.73–8.22 (m, 9H, Ar-H), 8.87 (s, 1H, OH), and 9.30 (s, 1H, NH); ^13^C NMR (100 MHz, CDCl_3_); δ 113.3, 113.8, 114.9, 120.6, 123.2, 123.3, 126.5, 127.4, 129.7, 130.6, 134.3, 134.8, 135.7, 156.4, 159.1, 164.6, and 166.8. MS: 357 (M + 1); Anal. Calcd. For C_20_H_11_N_3_O_4_: C, 67.23; H, 3.10; N, 11.76; Found: C, 66.92; H, 2.81; N, 11.74%.

4-(3,4-Dichloro-2-hydroxyphenyl)-6-(1,3-dioxoisoindolin-2-yl)-2-oxo-1,2-dihydropyridine-3-carbonitrile **8d**.

Dark brown powder; yield: 79%; m.p. 233–235 °C. FT-IR: 3446 (OH), 3255 (NH), 3090, 2921 (CH), 2214 (CN), 1680 (C=O), and 1597 (C=N); ^1^H NMR (400 MHz, CDCl_3_); δ 6.62–8.23 (m, 7H, Ar-H), 9.24 (s, 1H, OH), and 9.34 (s, 1H, NH); ^13^C NMR (100 MHz, CDCl_3_); δ 112.1, 113.4, 116.8, 121.1, 123.2, 124.5, 126.7, 128.2, 130.3, 131.4, 133.2, 134.5, 137.4, 152.3, 157.7, 161.9, and 164.7. MS: 426 (M + 1); Anal. Calcd. For C_20_H_9_Cl_2_N_3_O_4_: C, 56.36; H, 2.13; N, 9.86; Found: C, 56.15; H, 2.04; N, 9.34%.

### 3.3. Biological Assays

#### 3.3.1. Antioxidant Assays

The DPPH free radical, ABTS cation radical, and superoxide anion radical scavenging approaches were used in the antioxidant activity of the present study. The capacity for the radical scavenging was represented by EC_50_ values [56]. The EC_50_ values for the aforementioned antioxidant activities show the proportion of free, cation, and anion radicals that the tested samples were able to scavenge at 50% [57,58]. Every sample was observed three times. Butylated hydroxyanisole (BHA) is the positive control drug in this evaluation of antioxidant activity [59].

#### 3.3.2. Inhibition of α-Amylase and α-Glycosidase

α-Amylase (EC 3.2.1.1, type-VI B porcine pancreatic α-amylase) and α-glycosidase (EC 3.2.1.20, type-1 α-glycosidase) inhibition was tested using soluble starch (1%) and pNPG, respectively, according to the modified method described by Ramu [60,61], with acarbose as a positive control. Inhibitory activity of α-amylase and α-glycosidase was expressed as percent inhibition, calculated using the following formula.
Inhibition (%) = (A _control_ − A _sample_)/A _control_ × 100

The IC_50_ values were obtained from a graph that correlated the percentage of the inhibition of each sample with its concentration. Each experiment included three replicates, along with appropriate blanks. The concentration needed to reduce α-glycosidase activity by 50% under the specified assay conditions was referred to as the IC_50_ [62,63,64].

#### 3.3.3. Kinetics of α-Glycosidase and α-Amylase Inhibition

The enzyme kinetics of the compounds’ inhibition of the glycosidase activity were examined using different concentrations of the substrate against the IC_20_, IC_40_, and IC_60_ inhibitory concentrations of the compounds, as previously reported [42,65,66,67]. The type of inhibition, Km, and Vmax were calculated using a double-reciprocal plot [68] of the substrate concentration and velocity (1/V against 1/[pNPG]). The inhibitory constant (Ki) was also determined using the Dixon plot [69].

### 3.4. Computational Studies

#### 3.4.1. Molecular Docking Simulation

The homology model of α-glycosidase was built from the *Saccharomyces cerevisiae* (yeast) α-glycosidase MAL-32 sequence obtained from UniProt (UniProt ID: P38158), using the SWISS-MODEL (https://swissmodel.expasy.org/) (accessed on: 16 June 2023). Validation of the protein model was completed in previous studies and was found to be stable with the X-ray crystal structure of *S. cerevisiae* isomaltase (PDB ID: 3AXH), showing a 72% identical and 84% similar sequence [45,46], whereas the X-ray crystal structure of porcine pancreatic α-amylase (PDB ID: 1DHK) was retrieved from the RCSB PDB database. Since human α-glycosidase is not yet characterized, the authors had to use a *S. cerevisiae*/yeast model in both in vitro and in silico studies [70,71,72]. The 2D structures of the quinoline and isoindoline–polycyclic compounds were drawn and 3D optimized using ACD ChemSketch. Acarbose was chosen as the control drug for the study both in vitro and in silico, and its structure was retrieved from the PubChem database (https://pubchem.ncbi.nlm.nih.gov/) (accessed on: 16 June 2023).

According to previous works [73,74], both the protein and ligand structures were constructed for molecular docking simulation using AutoDock Tools 1.5.6. Water and heteroatoms were removed from the protein structures. Polar hydrogens, on the other hand, were used to stabilize it. Using Kollmann-united charges and Gasteiger charges, the energy of the protein and ligand structures was reduced. Prior to obtaining the completed protein and ligand structures in PDBQT format for the molecular docking simulation, all atoms were assigned an AutoDock 4 atom type after energy reduction [75].

The selection of the binding site for α-glycosidase was performed according to a previous study [39], whereas for α-amylase, it was performed using literature analysis [49]. Using AutoDock Tools 1.5.6, the binding pocket of the proteins was set in the grid box. Details of the grid boxes for both α-glycosidase and α-amylase have been given in Table 9.

The compounds’ molecular docking was accomplished using command-line software known as AutoDock Vina 1.1.2. It perturbs and allocates ligands into the target site using the Broyden–Fletcher–Goldfarb–Shanno (BGFS) algorithm, and it analyzes the scoring function of each ligand conformation [76]. Because of the huge number of torsions induced during ligand synthesis, ligands were treated as flexible throughout the docking simulation, whereas the protein was assumed to be rigid. However, ligand molecules were permitted 10 degrees of freedom. The initial binding posture with a zero root-mean-square deviation (RMSD) of atomic positions is regarded to be extremely genuine [77].

Furthermore, they have the highest binding affinity of any position, indicating that the binding is more effective. Biovia Discovery Studio Visualizer 2021, an open-source visualizing GUI program, was used to complete the visualization of the molecular docking simulation. Binding affinity, total number of bonds, and respective hydrogen bonds were used to estimate the amount of ligand interaction [78].

#### 3.4.2. Molecular Dynamics Simulation

Docked complexes of α-glycosidase and α-amylase complexed with **7d** and acarbose were used for the molecular dynamics (MD) simulation. According to a previous study [79], the MD simulation was undertaken using the biomolecular software program GROMACS-2018.1. GROMACS is a comprehensive software program for modeling Newtonian equations of motion or performing molecular dynamics on systems involving hundreds to millions of particles. It is designed particularly for biological compounds having several complicated binding connections, such as proteins, lipids, and nucleic acids. The software excels at calculating nonbonded interactions, which are frequently regarded as the most critical in simulations [80]. The SwissParam server was utilized to construct the ligand topology, and the CHARMM36 forcefield was used to approximate the ligand structures. On the other hand, protein structure was added with the CHARMM36 forcefield using the pdb2gmx module. The next stage was to perform 5000 steps of energy reduction in a vacuum using the steepest descent technique [81].

Each protein complex was separated from the box’s borders by a distance of ten. The solvent was integrated into the TIP3P water model with the appropriate amount of Na+ and Cl- counterions to maintain the required 0.15 M salt content. Six simulations were performed for 100 ns at a 310 K temperature and 1 bar pressure. The trajectory analysis of the RMSD, RMSF, Rg, SASA, and ligand hydrogen bond parameters was performed, and the results were plotted in graphical form using XMGRACE v1.0, a GUI-based software used for plotting MD simulation results [79,80,81].

#### 3.4.3. Binding Free Energy Calculations

The MD simulation results for α-glycosidase and α-amylase complexed with **7d** and acarbose were used to calculate binding free energy using the Molecular Mechanics/Poisson–Boltzmann Surface Area (MM-PBSA) approach. It is another use of molecular dynamics simulations and thermodynamics to determine the amount of ligand binding to the protein. The binding free energy for each ligand–protein combination was calculated using the g_mmpbsa v1.2 software with the MmPbSaStat.py script and the GROMACS 2018.1 trajectories as the input [47,48,49,50]. The binding free energy is calculated using three components in the g_mmpbsa program: molecular mechanical energy, polar and apolar solvation energies, and molecular mechanical energy. The calculation is performed utilizing the MD trajectories from the last 50 ns to determine G with dt 1000 frames. It is evaluated using molecular mechanical energy and polar–apolar solvation energies. Equations (1) and (2), used to calculate the free binding energy, are given below [51,52].
ΔG_Binding_ = G_Complex_ − (G_Protein_ + G_Ligand_) (1)
ΔG = ΔE_MM_ + ΔG_Solvation_ − TΔS = ΔE_(Bonded+non-bonded)_ + ΔG_(Polar+non-polar)_ – TΔS (2)

G_Binding_: binding free energy; G_Complex_: total free energy of the protein–ligand complex; G_Protein_ and G_Ligand_: total free energies of the isolated protein and ligand in a solvent, respectively; ΔG: standard free energy; ΔE_MM_: average molecular mechanics potential energy in a vacuum; G_Solvation_: solvation energy; ΔE: the total energy of bonded as well as nonbonded interactions; ΔS: change in the entropy of the system upon ligand binding; T: temperature in Kelvin [53,54,55].

#### 3.4.4. Physicochemical and ADMET Properties

The physicochemical and biological absorption, distribution, metabolism, excretion, and toxicity (ADMET) properties of the selected compound were performed using the ADMETlab 2.0 server (https://admetmesh.scbdd.com/) (accessed on: 16 June 2023). The SMILES format of compound **7d** was obtained and subjected to the calculation of both the abovementioned properties. ADMETlab 2.0 is designed to provide a systematic evaluation of the ADMET properties, as well as physicochemical properties and medicinal chemistry friendliness. It incorporates updated functional modules, predictive models, explanations, and an improved user interface. By leveraging these enhancements, ADMETlab 2.0 aims to assist medicinal chemists in accelerating the drug research and development process [73,74].

### 3.5. Statistical Analyses

The experiments were carried out in triplicate, and the findings were reported as mean ± SE. The EC_50_ values were calculated using the Graph Pad PRISM software (version 4.03) and were considered statistically significant if the ‘*p*’ values were 0.05 or below. Results were subjected to a one-way analysis of variance (ANOVA), followed by Duncan’s multiple range test, and the mean comparisons were performed by Duncan’s multiple range test using SPSS (version 21.0, Chicago, IL, USA).

## 4. Conclusions

In conclusion, we have successfully synthesized a series of polycyclic compounds containing different heterocycles, such as tetrazole, triazole, pyrazole, and pyridine. These compounds were prepared using various synthetic approaches, including one-pot three-component condensation reactions. The researchers also demonstrated the synthesis of Schiff bases and pyrazoles through specific reactions. In the antidiabetic assays, one compound, namely **7d**, stood out as a potential lead candidate. Compound **7d** showed superior inhibition of both α-glycosidase (IC_50_: 0.07 mM) and α-amylase (0.21 mM) compared to the standard acarbose, indicating its potential as an effective antidiabetic agent. It also showed the potent antioxidant activity of EC_50_ = 0.65, 0.52, and 0.93 mM during the free radical scavenging assays (DPPH, ABTS, and superoxide anion radicals). Molecular docking (α-glycosidase: −10.9 kcal/mol and α-amylase: −9.0 kcal/mol) and dynamics simulations further confirmed the binding interactions and stability of **7d** with the target enzymes, providing valuable insights into its mode of action. Additionally, physicochemical and ADMET studies were conducted, which revealed that compound **7d** exhibited no toxic effects. This finding suggests that **7d** could be considered for further development as an oral antidiabetic medication. To summarize, the combination of in silico and in vitro studies has yielded promising results for compound **7d** as a potential antidiabetic agent. Its efficient inhibition of key enzymes, stable binding interactions, lack of toxicity, and oral availability make it a promising candidate for further investigation during in vivo and clinical trials, with the need for prior dose-dependent in vivo toxicity experiments. These findings open up new possibilities for the development of novel therapeutics in the field of diabetes treatment. However, further studies and validations are required to fully assess the safety and efficacy of compound **7d** before it can be considered for human trials.

## Data Availability

Data are available in the article and the Appendix A.

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
