# Peer review of "Quinoline- and Isoindoline-Integrated Polycyclic Compounds as Antioxidant, and Antidiabetic Agents Targeting the Dual Inhibition of α-Glycosidase and α-Amylase Enzymes"

_pharmaceuticals, 2023, doi:10.3390/ph16091222_

Round 1

Reviewer 1 Report

The following is to be addressed before the manuscript gets accepted

1. Too long sentences to be avoided in the introduction

2. Numbering of figures should be ascending, figure 6 missing

3. Table 3, linking z (on footnote) is missing on the table

4. Reference 21 is missing

5. References should be in ascending order, reference 45 comes after 46

6. All the figures should be converted to high quality images

7. Some of the references are worth citing under 

a. Antioxidant activity (section 3.3.1): https://doi.org/10.1002/jhet.4699

 b.  Section 3.3.2

https://doi.org/10.1080/10406638.2023.2169471,https://doi.org/10.3390/ph15101250, https://doi.org/10.1016/j.ejmech.2023.115549; 

c. Like Isoindoline-pyridine you can compare other fused systems like pyridine-imidazoles example: https://doi.org/10.1016/j.ejps.2022.106323

8. Discussion part is missing, needs to be included. 

9. Conclusion part: rather focusing on the observation, inference should be highlighted

10. Supporting information images should be of high quality

The manuscript can be accepted after the authors revision

Author Response

RESPONSE TO REVIEWER COMMENTS

We thank the reviewer for providing his valuable suggestions on improvement of this manuscript. The suggestions were considered and the same were incorporated during the revision of this of manuscript wherever applicable. A detailed response to the reviewer comments were given below.

The following is to be addressed before the manuscript gets accepted

  1. Too long sentences to be avoided in the introduction

Authors’ Response: According to the reviewer’s suggestion, long sentences have been shortened in the introduction.

  1. Numbering of figures should be ascending, figure 6 missing.

Authors’ Response: According to the reviewer’s suggestion, all the figures have been cited in an ascending pattern including figure 6.

  1. Table 3, linking z (on footnote) is missing on the table

Authors’ Response: According to the reviewer’s suggestion, linking z (on footnote) has been completed.

  1. Reference 21 is missing

Authors’ Response: According to the reviewer’s suggestion, the missing reference 21 has now been added.

  1. References should be in ascending order, reference 45 comes after 46

Authors’ Response: According to the reviewer’s suggestion, the references have been arranged in an increasing order, however to match their relevance with the text few of them have been repeated in the text.

  1. All the figures should be converted to high quality images

Authors’ Response: According to the journal’s requirement, the authors have provided the best resolution (1200 dpi) for every image. All the images have been provided are original and are obtained from softwares used for in silico experiments. Authors have not altered the images since it leads to the manipulation of the original scientific data. Therefore, the authors request the reviewer to accept the images and proceed for the future proceedings.

  1. Some of the references are worth citing under 
  2. Antioxidant activity (section 3.3.1): https://doi.org/10.1002/jhet.4699
  3.  Section 3.3.2

https://doi.org/10.1080/10406638.2023.2169471,https://doi.org/10.3390/ph15101250, https://doi.org/10.1016/j.ejmech.2023.115549; 

  1. Like Isoindoline-pyridine you can compare other fused systems like pyridine-imidazoles example: https://doi.org/10.1016/j.ejps.2022.106323

Authors’ Response: According to the reviewer’s suggestion, the suggested references have been cited.

  1. Discussion part is missing, needs to be included. 

Authors’ Response: According to the reviewer’s suggestion, Results and Discussion section has been modified. Authors apologize for the typing error during the manuscript preparation.

  1. Conclusion part: rather focusing on the observation, inference should be highlighted

Authors’ Response: According to the reviewer’s suggestion, inference has been highlighted in the new conclusion section.

  1. Supporting information images should be of high quality

Authors’ Response: According to the reviewer’s suggestion, Supporting information images have been provided in high quality.

Reviewer 2 Report

This manuscript provides a comprehensive discussion on the antioxidant and antidiabetic properties of quinoline and isoindoline integrated polycyclic compounds, targeting dual inhibition of α-glycosidase and α-amylase enzymes. From an organic chemistry perspective, the primary concern is the absence of 1H and 13C NMR spectra of all synthesized compounds in the Supporting Information, which are essential for acceptance. In general, the manuscript meets the required criteria for publication in 'Pharmaceuticals' after minor revision. Nonetheless, there are a few specific issues that require attention:  

(1) See the abstract section. The most relevant biological activity values could be included.  

(2) See the introduction section. Certain biological activity values could be incorporated to support specific statements.

(3) See Schemes 1, 2, and 3. The yields of all synthesized compounds should be included in the Schemes. Additionally, a discussion on the yields could be added to section 2.1. Chemistry.

(4) See Tables 1 and 2. Specify the number of repetitions at the end of each table to determine the mean SE.

(5) See the Figure 2. The resolution should be improved.

(6) See 2.3.1. Molecular docking simulation. Specify the PDB ID code for each enzyme in parentheses

(7) See Figures 3 and 4. Consider enhancing the resolution. Alternatively, it may be split into two parts for better visualization.

(8) See Figures 5 and 6. Consider enhancing the resolution.

(9) See 3.2. Synthesis. I kindly request a thorough revision of the NMR data for each synthesized compound (i.e. number of protons and coupling constants).

(10) See 3.2. Synthesis. It is recommended to include the frequency of the spectrometer and the type of deuterated solvent used, such as 1H NMR (400 MHz, CDCl3) or 13C NMR (101 MHz, CDCl3).

(11) See the conclusion section. It would be beneficial to include the most significant biological activity values to bolster specific statements.

(12) See the references. The DOI of each article is required.

(13) Please refer to the Supporting Information for the inclusion of 1H and 13C NMR spectra of all compounds. Specifically, the 1H NMR spectra should be processed to include the integration of each proton signal and calibrated with the residual signal of the deuterated solvent. Additionally, the presence of impurities (<5%) can be detected using 1H NMR.  

Author Response

RESPONSE TO REVIEWER COMMENTS

We thank the reviewer for providing his valuable suggestions on improvement of this manuscript. The suggestions were considered and the same were incorporated during the revision of this of manuscript wherever applicable. A detailed response to the reviewer comments were given below.

Comments

 (1) See the abstract section. The most relevant biological activity values could be included.  

Authors’ Response: According to the reviewer’s suggestion, the abstract has been included with relevant outcomes of biological activity values.

(2) See the introduction section. Certain biological activity values could be incorporated to support specific statements.

Authors’ Response: According to the reviewer’s suggestion, biological activity values have been incorporated to support statements about various pharmacological activities of the compounds.

(3) See Schemes 1, 2, and 3. The yields of all synthesized compounds should be included in the Schemes. Additionally, a discussion on the yields could be added to section 2.1. Chemistry.

Authors’ Response: According to the reviewer’s suggestion, the yields of all synthesized compounds have included in the Schemes and discussion section.

(4) See Tables 1 and 2. Specify the number of repetitions at the end of each table to determine the mean SE.

Authors’ Response: The authors appreciate the reviewer’s concern about the observations provided in Table 1 and Table 2. However, since the authors have followed Duncan's multiple range test approach define the mean SE, the number of repetitions at the end of each table would be immensely high to provide in the given space. Therefore the authors request the reviewer to consider the observations in the current format and proceed for future proceedings.

(5) See the Figure 2. The resolution should be improved.

Authors’ Response: According to the journal’s requirement, the authors have provided the best resolution (1200 dpi) for every image. All the images have been provided are original and are obtained from softwares used for in silico experiments. Authors have not altered the images since it leads to the manipulation of the original scientific data. Therefore, the authors request the reviewer to accept the images and proceed for the future proceedings.

(6) See 2.3.1. Molecular docking simulation. Specify the PDB ID code for each enzyme in parentheses

Authors’ Response: According to the reviewer’s suggestion, the PDB ID code for each enzyme in parentheses has been included.

(7) See Figures 3 and 4. Consider enhancing the resolution. Alternatively, it may be split into two parts for better visualization.

Authors’ Response: According to the journal’s requirement, the authors have provided the best resolution (1200 dpi) for every image. All the images have been provided are original and are obtained from softwares used for in silico experiments. Authors have not altered the images since it leads to the manipulation of the original scientific data. Therefore, the authors request the reviewer to accept the images and proceed for the future proceedings.

Also, the authors request the reviewer to retain the images in one piece to avoid confusion to the readers between 3D and 2D images. The 3D and 2D images have been kept together because 3D images depict the specific rotation of the compound in the binding site and 2D images depict the binding interaction.

(8) See Figures 5 and 6. Consider enhancing the resolution.

Authors’ Response: According to the journal’s requirement, the authors have provided the best resolution (1200 dpi) for every image. All the images have been provided are original and are obtained from softwares used for in silico experiments. Authors have not altered the images since it leads to the manipulation of the original scientific data. Therefore, the authors request the reviewer to accept the images and proceed for the future proceedings.

(9) See 3.2. Synthesis. I kindly request a thorough revision of the NMR data for each synthesized compound (i.e. number of protons and coupling constants).

Authors’ Response: Thank you for the reviewer, we have revised the NMR data and checked the number of number of protons well, but we couldn't able to count the coupling constant, due the only aromatic proton can be count and their signals close to each others.

(10) See 3.2. Synthesis. It is recommended to include the frequency of the spectrometer and the type of deuterated solvent used, such as 1H NMR (400 MHz, CDCl3) or 13C NMR (101 MHz, CDCl3).

Authors’ Response: According to the reviewer’s suggestion, the frequency of the spectrometer and the type of deuterated solvent used were added.

(11) See the conclusion section. It would be beneficial to include the most significant biological activity values to bolster specific statements.

Authors’ Response: According to the reviewer’s suggestion, the most significant biological activity values have been included in the new conclusion section.

(12) See the references. The DOI of each article is required.

Authors’ Response: According to the reviewer’s suggestion, DOI of each article has been added.

(13) Please refer to the Supporting Information for the inclusion of 1H and 13C NMR spectra of all compounds. Specifically, the 1H NMR spectra should be processed to include the integration of each proton signal and calibrated with the residual signal of the deuterated solvent. Additionally, the presence of impurities (<5%) can be detected using 1H NMR. 

Authors’ Response: Thank you for the reviewer, the integration of proton signal in 1 H NMR for the most of the targeted compounds have marked in the supporting information.